# Correlations between Genetic Polymorphisms in Long Non-Coding RNA *PRNCR1* and Gastric Cancer Risk in a Korean Population

**DOI:** 10.3390/ijms20133355

**Published:** 2019-07-08

**Authors:** Jang Hee Hong, Eun-Heui Jin, Hyojin Kang, In Ae Chang, Sang-Il Lee, Jae Kyu Sung

**Affiliations:** 1Clinical Trials Center, Chungnam National University Hospital, Daejeon 35015, Korea; 2Department of Pharmacology, Chungnam National University College of Medicine, Daejeon 35015, Korea; 3Research Institute for Medical Sciences, Chungnam National University College of Medicine, Daejeon 35015, Korea; 4Department of Surgery, Chungnam National University Hospital, Chungnam National University College of Medicine, Daejeon 35015, Korea; 5Department of Internal Medicine, Chungnam National University Hospital, Chungnam National University College of Medicine, Daejeon 35015, Korea

**Keywords:** *PRNCR1*, polymorphism, gastric cancer, intestinal type, lymph node metastasis

## Abstract

We evaluated the association between prostate cancer non-coding RNA 1 (*PRNCR1*) polymorphisms and the risk of developing gastric cancer (GC) and GC subgroups in Korea. A case–control study was conducted with 437 GC patients and 357 healthy controls using a TaqMan genotyping assay. A chi-squared test, binary logistic regression, and genetic models were used to explore the association between five *PRNCR1* polymorphisms and GC risk. After adjusting for gender and age, overall analyses using the recessive model indicated that the rs13252298 GG genotype was significantly associated with increased risk of intestinal-type gastric cancer (IGC). In the stratification analyses, the recessive model indicated that the rs1016343 TT genotype was significantly associated with decreased GC risk in individuals aged <60 years showing lymph node metastasis (LNM)-negative results. The rs13252298 GG genotype in the recessive model showed increased GC risk in subjects aged ≥60 years showing LNM-positive results and those aged ≥60 years in tumor stage III. In the dominant model, the rs16901946 combined genotype (AG/GG) was significantly associated with increased GC risk in subjects aged <60 years with tumor stage III. In the recessive model, the rs16901946 GG genotype was associated with decreased risk of GC and IGC in males aged ≥60 years. Thus, genetic variations in *PRNCR1* may contribute to susceptibility to GC.

## 1. Introduction

Gastric cancer (GC) is one of the most common forms of cancer worldwide. Despite a steady decline in GC incidence and mortality rates over the past few decades, the rates are still high in Asian countries. According to a report by the Korean National Cancer Center, GC is the third most common cancer, with 25,872 new cases and 7138 deaths recorded in Korea in 2016 [1,2,3]. 

Approximately 80% of disease-related polymorphisms occur in non-coding regions consisting of introns and intergenic regions [4]. Genome-wide association studies have reported that a large number of polymorphisms are associated with cancer and that these cancer-related polymorphisms are associated with long non-coding RNAs (lncRNAs) [5,6,7]. LncRNAs are non-translated RNA molecules over 200 nucleotides in length. Recently, it was reported that lncRNAs are involved in tumorigenesis [8,9]. Moreover, germline variants can affect lncRNA expression, regulating cancer development and progression [10]. Indeed, several studies have demonstrated that lncRNA genetic variants are related to susceptibility of various cancers, including breast cancer [11,12], colon cancer [13], gastric cancer [14,15], lung cancer [16], and prostate cancer [10,17]. Particularly, lncRNA prostate cancer non-coding RNA 1 (*PRNCR1*), transcribed from a non-coding region of chromosome 8q24, is involved in the carcinogenesis of prostate cancer (PC) through activation of the androgen receptor [18], and lncRNA *PRNCR1* polymorphisms have been correlated with various cancers, including GC [14,18,19,20,21,22,23]. 

Based on previous findings, we hypothesized that polymorphisms in the lncRNA *PRNCR1* might affect genetic susceptibility to GC. Therefore, we conducted a case–control study to elucidate the association between single nucleotide polymorphisms (SNPs) in *PRNCR1* and risk of developing GC in a Korean population. We further analyzed the impact of *PRNCR1* polymorphisms on GC risk in combination with various characteristics and clinical features, including age, sex, tumor differentiation, histologic type, T classification, lymph node metastasis (LNM), and tumor stage.

## 2. Results

### 2.1. Patient Characteristics and Single Nucleotide Polymorphisms (SNP) Selection

The characteristics and clinical features of the 437 patients with GC and the 363 cancer-free controls are presented in Table 1. There was a significant difference in the age and gender distribution between the two groups (*p* < 0.001 and *p* < 0.001, respectively). The mean age was 65.3 ± 11.1 years for GC patients and 58.1 ± 8.9 years for the controls. The proportion of male subjects was significantly higher in the group with GC (69.6%), whereas the number of female subjects was higher in the control group (67.2%). Of the 437 patients with diagnosed GC, more than half were classified as having the intestinal type (55.8%), making it the most common type, followed by the diffuse-type and the mixed-type. The majority of the patients did not show lymph node metastasis (LNM) (60.9%) and were classified as T1 (50.6%) and tumor stage I (58.8%). We selected five *PRNCR1* SNPs: rs1016343, rs13252298, rs7841060, rs16901946, and rs1456315, which have been previously associated with cancer. 

### 2.2. Associations Between PRNCR1 SNPs and GC risk

The genotype frequencies of rs1016343, rs13252298, rs7841060, and rs16901946 were in Hardy–Weinberg equilibrium (HWE) for both the GC group (*p* = 0.772, *p* = 0.968, *p* = 0.668, and *p* = 0.821, respectively) and the control group (*p* = 0.591, *p* = 0.143, *p* = 0.610, and *p* = 0.978, respectively) (Table 2). However, rs1456315 frequencies were not in HWE for either GC or controls (*p* < 0.05). The rs1456315 was, therefore, excluded from the genotype analysis because of Hardy-Weinberg disequilibrium. LD coefficients (|D’|) were estimated among the four SNPs, and an absolute LD (|D’| = 1 and *r*^2^) was not found for any pair-wise combination among four SNPs using Haploview 4.0 software. To determine whether rs1016343, rs13252298, rs7841060, and rs16901946 were associated with a higher risk of GC or GC subgroups, we compared the genotypic frequencies between the GC group and the control group. After adjusting for age and gender, the recessive model indicated that the rs13252298 GG genotype was associated with an increased risk of intestinal-type gastric cancer (IGC) (OR = 1.92, 95% CI = 1.01–3.63, *p* = 0.045), as compared to the rs13252298 AA/AG genotypes; the remaining SNPs showed no significant associations (Table 2).

### 2.3. Stratified Analysis for Four PRNCR1 SNPs

To further estimate the possible correlation between the four SNPs and GC risk in GC subgroups, we performed stratified analyses based on various patient characteristics, including age, sex, LNM, T classification, and tumor stage. As shown in Table 3, after adjusting for age and gender, our recessive model demonstrated that the rs1016343 TT genotype was significantly associated with a decreased risk of GC in subjects aged <60 years showing LNM-negative results (odds ratio, OR = 0.29, 95% confidence interval, CI = 0.09–0.94, *p* = 0.038), when compared to the rs1016343 CC/CT genotypes. Moreover, according to the recessive model, the rs13252298 GG genotype was associated with an increased risk of GC for those aged ≥60 years showing LNM-positive results (OR = 2.80, 95% CI = 1.15–6.82, *p* = 0.024) and those aged ≥60 years in tumor stage III (OR = 3.39, 95% CI = 1.35–8.52, *p* = 0.009), as compared to the rs13252298 AA/AG genotypes. Furthermore, the dominant model showed that the rs16901946 combined genotype (AG/GG) had a significant association with increased risk of GC in subjects aged <60 years in tumor stage III (OR = 2.38, 95% CI = 1.15–4.94, *p* = 0.020). The recessive model also showed that the rs16901946 GG genotype was associated with a decreased risk of GC (OR = 0.43, 95% CI = 0.19–0.98, *p* = 0.046) and IGC (OR = 0.38, 95% CI = 0.16–0.89, *p* = 0.026) in male subjects aged ≥60 years when compared to the rs16901946 AA/AG genotype. However, the association between polymorphisms in *PRNCR1* and tumor differentiation was not observed.

## 3. Discussion

To date, most studies, including GWAS and a meta-analysis, have reported on the association between genetic variations in *PRNCR1* and PC [18,19,20,21]. However, little is known about the associations between *PRNCR1* polymorphisms and GC. Furthermore, results thus far have been inconsistent, and the genotyping methods have also varied [14,23]. In this case–control study, we investigated the association between lncRNA *PRNCR1* polymorphisms and GC risk in a Korean population using a reliable TaqMan genotyping assay. We found that the rs13252298 GG genotype was associated with 1.92-times increased risk of IGC. Li et al. previously reported that there was an association between the rs13252298 AG genotype and GC in the Chinese population; in contrast, He et al. reported a lack of association between the rs13252298 polymorphism and GC in the Chinese population [14,23]. Interestingly, our age-stratified analysis found that the rs1016343 TT genotype was associated with 0.92-times reduced risk of GC in subjects aged <60 years showing LNM-negative results; the rs13252298 GG genotype was associated with 2.80- and 3.39-times increased risk of GC in subjects aged ≥60 years showing LNM-positive results and aged ≥60 years in tumor stage III, respectively. The rs16901946 AG/GG genotype was associated with 2.38-times increased risk of GC in subjects <60 years in tumor stage III. However, He et al. described an association between the rs16901946 genotype and increased risk of GC in younger and tumor stage I+II subjects. Interestingly, in our study the rs1016343 TT genotype was associated with 0.29-times decreased risk of GC in those aged <60 years showing LNM-negative results. Li et al. demonstrated an association between the rs1456315 GG genotype and a decreased risk of GC. However, we could not analyze an association because of the Hardy–Weinberg disequilibrium in either the GC group or the control group (*p* < 0.05).

There are several limitations to our study. First, the sample size was too small to have statistical power for our stratified analyses. Second, although *Helicobacter pylori* is an independent risk factor [24,25], we did not investigate its relevance with regard to the *PRNCR1* polymorphisms in GC risk because of ethical considerations. Third, we failed to explore whether there is an association between the genetic factors and smoking, drinking, and diet related to GC risk owing to lack of these data from the GC and control groups. Fourth, our study was performed in a Korean population. Thus, we should include other ethnic groups and a larger sample size to confirm our results. Future studies will require that we assess the influence of these factors on GC.

In conclusion, our study suggests that the *PRNCR1* rs13252298 and rs16901946 polymorphisms are associated with increased GC risk and may exacerbate the development of GC. Moreover, the *PRNCR1* rs1016343 polymorphism may contribute to a decreased risk of GC in those aged <60 years showing LNM-negative results. Further studies are needed to validate our results in a larger population as well as in different ethnic groups.

## 4. Materials and Methods

### 4.1. Ethics Statement

The present study was conducted in accordance with the Declaration of Helsinki and was reviewed and approved by the Ethics Committee of the institutional review board of Chungnam National University Hospital on 23 July 2017. Informed consent was provided by all subjects when they were enrolled.

### 4.2. Study Subjects

In total, 437 GC patients and 357 healthy controls were enrolled in this study. The blood samples used in this study were provided by the Chungnam National Hospital Biobank, a member of the National Biobank of Korea, which is supported and audited by the Ministry of Health and Welfare of Korea. All individuals enrolled in this study provided their written informed consent for blood collection and use. GC patients were recruited from the outpatient clinic at the Chungnam National University Hospital, and classified according to the Lauren’s classification [26]. The subjects for the control group were randomly selected among healthy volunteers visiting the Chungnam National University Hospital medical center for their annual physical examinations; only individuals who had no history of cancer were included. 

### 4.3. DNA Isolation and Genotyping

Genomic DNA was isolated from the peripheral blood using the QIAamp DNA Blood Mini Kit (Qiagen GmbH, Hilden, Germany), according to the manufacturer’s instructions. Five SNPs (rs1016343, rs13252298, rs7841060, rs16901946, and rs1456315) in *PRNCR1* were selected based on previous reports [14,19,20,23,24,25] and genotyped using the Applied Biosystems TaqMan SNP Genotyping Assay with the StepOnePlus Real-time PCR System (Applied Biosystems, Foster City, CA, USA).

### 4.4. Statistical Analysis

Hardy–Weinberg equilibrium (HWE) for each SNP in the control groups was evaluated using the chi-squared test. A pair-wise comparison of biellelic loci was employed for the analyses of linkage disequilibrium (LD) using Haploview software version 4.0 (the Broad Institute, Cambridge, MA, USA). Differences in age and gender between the GC and control groups were calculated using the two-sided Pearson chi-squared test and the Mann–Whitney *U*-test. Two genetic models (dominant and recessive models) were used to analyze the associations. A binary logistic regression was used to estimate the GC risk according to odds ratios (ORs) and 95% confidence intervals (CIs). The association analysis was adjusted by age and sex, which were included in the model as covariates. All statistical analyses were performed using the SPSS (SPSS Inc., Chicago, IL, USA), version 20.0 for Windows. *p* <0.05 was considered statistically significant. 

## Figures and Tables

**Table 1 ijms-20-03355-t001:** Characteristics and clinical features of the gastric cancer group and the control group.

Variables	Gastric Cancers	Controls	*p*
	*N* (%)	*N* (%)	
Age (years) (mean ± SD)	437 (65.3 ± 11.1)	357 (58.1 ± 8.9)	<0.001 *
<60	185 (42.3)	186 (52.1)	0.007 ^†^
≥60	252 (57.7)	171 (47.9)	
Gender (%)			
Male	304 (69.6)	117 (32.8)	<0.001 ^†^
Female	133 (30.4)	240 (67.2)	
Tumor differentiation			
Differentiated	208 (47.6)		
Undifferentiated	190 (43.5)		
Missing	39 (8.9)		
Histological type (%)			
Intestinal	244 (55.8)		
Diffuse	140 (32.1)		
Mixed	53 (12.1)		
T classification (%)			
T1	221 (50.6)		
T2	59 (13.5)		
T3	16 (3.6)		
T4	141 (32.3)		
Lymph node metastasis (%)			
Negative	266 (60.9)		
Positive	171 (39.1)		
Tumor stage (%)			
I (A+B)	257 (58.8)		
II (A+B)	50 (11.5)		
III (A+B+C)	130 (29.7)		

SD, standard deviation. * Mann–Whitney *U*-test. ^†^ Two-sided Pearson’s chi-squared test.

**Table 2 ijms-20-03355-t002:** Genotype and allelic frequencies of *PRNCR1* polymorphisms in subjects and their association with GC risk.

Genotype	CON	GC vs. CON			IGC vs. CON			DGC vs. CON		
	*N* (%)	*N* (%)	AOR (95% CI) ^a^	*p*	*N* (%)	AOR (95% CI) ^a^	*p*	N (%)	AOR (95% CI) ^a^	*p*
rs1016343										
Codominant										
CC	171 (47.9)	209 (47.8)	1		119 (48.5)	1		62 (44.6)	1	0.661
CT	158 (44.3)	191 (43.7)	0.94 (0.69–1.29)	0.709	106 (43.3)	0.94 (0.64–1.38)	0.756	66 (47.5)	1.10 (0.72–1.68)	0.892
TT	28 (7.8)	37 (8.5)	0.88 (0.49–1.56)	0.654	20 (8.2)	0.84 (0.42–1.67)	0.612	11 (7.9)	0.95 (0.44–2.06)	
Dominant										
CC	171 (47.9)	209 (47.8)	1		119 (48.6)	1		62 (44.6)	1	0.727
CT + TT	186 (52.1)	228 (52.2)	0.93 (0.69–1.26)	0.645	126 (51.4)	0.92 (0.64–1.33)	0.672	77 (55.4)	1.08 (0.72–1.61)	
Recessive										
CC + CT	329 (92.2)	406 (91.5)	1		225 (91.8)	1		128 (92.1)	1	0.790
TT	28 (8.7)	37 (8.5)	0.90 (0.52–1.57)	0.718	20 (8.2)	0.86 (0.44–1.68)	0.659	11 (7.9)	0.90 (0.43–1.91)	
HWE	0.591	0.772								
**rs13252298**										
Codominant										
AA	158 (44.3)	214 (49.0)	1		122 (49.6)	1		67 (48.6)	1	0.450
AG	171 (47.9)	182 (41.6)	0.90 (0.66–1.24)	0.518	97 (39.4)	0.87 (0.59–1.29)	0.497	61 (44.2)	0.85 (0.56–1.30)	0.854
GG	28 (7.8)	41 (9.4)	1.36 (0.77–2.40)	0.285	27 (11.0)	1.80 (0.93–3.49)	0.084	10 (7.2)	0.93 (0.42–2.06)	
Dominant										
AA	158 (44.3)	214 (49.0)	1		122 (49.6)	1		67 (48.6)	1	0.469
AG + GG	199 (55.7)	223 (51.0)	0.96 (0.71–1.31)	0.805	124 (50.4)	0.99 (0.69–1.43)	0.954	71 (51.4)	0.86 (0.57–1.30)	
Recessive										
AA+AG	329 (92.2)	396 (90.6)	1		219 (89.0)	1		128 (92.8)	1	0.982
GG	28 (7.8)	41 (9.4)	1.43 (0.83–2.47)	0.193	27 (11.0)	1.92 (1.01–3.63)	**0.045**	10 (7.2)	1.01 (0.47–2.18)	
HWE	0.143	0.968								
rs7841060										
Codominant										
TT	169 (47.4)	204 (46.7)	1		116 (47.5)	1		61 (43.5)	1	0.556
TG	159 (44.5)	195 (44.6)	0.96 (0.70–1.31)	0.794	108 (44.3)	0.96 (0.65–1.40)	0.814	68 (48.6)	1.14 (0.75–1.73)	0.842
GG	29 (8.1)	38 (8.7)	0.88 (0.50–1.55)	0.647	20 (8.2)	0.82 (0.41–1.64)	0.578	11 (7.9)	0.92 (0.43–2.00)	
Dominant										
TT	169 (47.3)	204 (46.7)	1		116 (47.5)	1		61 (43.6)	1	0.643
TG + GG	188 (52.7)	233 (53.3)	0.95 (0.70–1.28)	0.716	128 (52.5)	0.93 (0.65–1.35)	0.711	79 (56.4)	1.10 (0.73–1.65)	
Recessive										
TT + TG	328 (91.9)	399 (91.3)	1		224 (91.8)	1		129 (92.1)	1	0.705
GG	29 (8.1)	38 (8.7)	0.89 (0.52–1.54)	0.688	20 (8.2)	0.84 (0.43–1.63)	0.609	11 (7.9)	0.87 (0.41–1.82)	
HWE	0.610	0.668								
rs16901946										
Codominant										
AA	178 (49.9)	208 (47.6)	1		117 (48.0)	1		68 (48.6)	1	0.506
AG	147 (41.1)	191 (43.7)	1.21 (0.88–1.66)	0.245	105 (43.0)	1.26 (0.85–1.85)	0.253	62 (44.3)	1.15 (0.76–1.76)	0.643
GG	32 (9.0)	38 (8.7)	0.84 (0.49–1.46)	0.539	22 (9.0)	0.78 (0.40–1.49)	0.445	10 (7.1)	0.83 (0.38–1.83)	
Dominant										
AA	178 (49.9)	208 (47.6)	1		117 (48.0)	1		68 (48.6)	1	0.658
AG + GG	179 (50.1)	229 (52.4)	1.13 (0.84–1.54)	0.414	127 (52.0)	1.15 (0.79–1.65)	0.468	72 (51.4)	1.10 (0.73–1.64)	
Recessive										
AA + AG	325 (91.0)	399 (91.3)	1		222 (91.0)	1		130 (92.9)	1	0.516
GG	32 (9.0)	38 (8.7)	0.77 (0.45–1.31)	0.338	22 (9.0)	0.70 (0.37–1.32)	0.269	10 (7.1)	0.78 (0.36–1.67)	
HWE	0.978	0.821								

CON, control; GC, gastric cancer; IGC, intestinal-type gastric cancer; DGC, diffuse-type gastric cancer; AOR, adjusted odds ratio; CI, confidence interval; HWE, Hardy–Weinberg equilibrium. ^a^ Adjusted for age and gender. The significant results are in bold.

**Table 3 ijms-20-03355-t003:** Stratified analysis of *PRNCR1* polymorphisms in GC patients and controls by age.

SNP	Variable	GC vs. CON							
		Dominant (ht+mt/wt)				Recessive (mt/wt+ht)			
		GC	CON	AOR (95% CI) ^a^	*p*	GC	CON	AOR (95% CI) ^a^	*p*
**rs1016343**	Gender (M)	76/56	28/17	0.93 (0.46–1.90)	0.849	8/124	6/39	0.43 (0.14–1.32)	0.140
Age	Gender (F)	27/28	68/73	1.00 (0.53–1.90)	0.992	2/53	9/132	0.40 (0.08–2.07)	0.276
<60	IGC	43/44	96/90	0.72 (0.38–1.37)	0.320	4/83	15/171	0.34 (0.10–1.19)	0.090
	DGC	47/30	96/90	1.43 (0.79–2.56)	0.236	4/73	15/171	0.50 (0.15–1.63)	0.248
	**LNM (−)**	73/50	96/90	1.28 (0.74–2.23)	0.380	5/118	15/171	0.29 (0.09–0.94)	**0.038**
	LNM (+)	30/34	96/90	0.71 (0.38–1.33)	0.284	5/59	15/171	0.76 (0.25–2.34)	0.634
	Tumor stage I + II	84/60	96/90	1.19 (0.70–2.02)	0.512	8/136	15/171	0.45 (0.17–1.24)	0.123
	Tumor stage III	19/24	96/90	0.61 (0.30–1.26)	0.184	2/41	15/171	0.40 (0.08–1.90)	0.248
≥60	Gender (M)	36/38	91/85	1.14 (0.66–1.97)	0.646	21/155	6/68	1.48 (0.56–3.88)	0.426
	Gender (F)	34/40	54/43	1.14 (0.66–1.97)	0.144	6/68	7/90	1.48 (0.56–3.88)	0.663
	IGC	83/75	90/81	1.09 (0.67–1.76)	0.727	16/142	13/158	1.39 (0.59–3.26)	0.453
	DGC	30/32	90/81	0.81 (0.45–1.48)	0.495	7/55	13/158	1.40 (0.51–3.84)	0.514
	LNM (−)	76/69	90/81	0.99 (0.62–1.59)	0.967	14/131	13/158	1.28 (0.55–3.01)	0.566
	LNM (+)	49/56	90/81	0.80 (0.47–1.36)	0.409	13/92	13/158	1.42 (0.57–3.53)	0.446
	Tumor stage I + II	83/83	90/81	0.92 (0.58–1.45)	0.706	17/149	13/158	1.40 (0.62–3.15)	0.418
	Tumor stage III	42/42	90/81	0.90 (0.51–1.58)	0.703	10/74	13/158	1.25 (0.47–3.37)	0.658
**rs13252298**	Gender (M)	69/63	27/18	0.75 (0.37–1.49)	0.405	8/124	1/44	2.69 (0.32–22.47)	0.360
Age	Gender (F)	33/21	89/53	0.88 (0.46–1.71)	0.716	6/48	16/126	0.95 (0.34–2.63)	0.916
<60	IGC	45/42	116/71	0.75 (0.40–1.40)	0.369	10/77	17/170	2.42 (0.83–7.00)	0.104
	DGC	44/32	116/71	0.82 (0.46–1.48)	0.517	4/72	17/170	0.78 (0.24–2.54)	0.680
	LNM (−)	67/54	116/71	0.86 (0.49–1.49)	0.590	10/111	17/170	1.26 (0.47–3.35)	0.645
	LNM (+)	35/30	116/71	0.72 (0.39–1.34)	0.297	4/61	17/170	1.13 (0.34–3.75)	0.847
	Tumor stage I + II	77/65	116/71	0.81 (0.48–1.37)	0.434	12/130	17/170	1.51 (0.60–3.81)	0.382
	Tumor stage III	25/19	116/71	0.82 (0.41–1.66)	0.585	2/42	17/170	0.69 (0.15–3.28)	0.645
≥60	Gender (M)	79/95	32/41	1.04 (0.59–1.81)	0.898	18/156	5/68	1.77 (0.62–5.06)	0.289
	Gender (F)	42/35	51/46	0.96 (0.50–1.87)	0.912	9/68	6/91	1.83 (0.54–6.21)	0.332
	IGC	79/80	83/87	1.07 (0.66–1.73)	0.780	17/142	11/159	1.73 (0.73–4.10)	0.210
	DGC	27/35	83/87	0.87 (0.47–1.59)	0.646	6/56	11/159	2.19 (0.74–6.48)	0.157
	LNM (−)	71/75	83/87	1.01 (0.63–1.62)	0.967	12/134	11/159	1.43 (0.58–3.56)	0.442
	**LNM (+)**	50/55	83/87	1.07 (0.62–1.82)	0.817	15/90	11/159	2.80 (1.15–6.82)	**0.024**
	Tumor stage I + II	80/86	83/87	1.01 (0.64–1.60)	0.975	14/152	11/159	1.37 (0.56–3.32)	0.493
	**Tumor stage III**	41/44	83/87	1.08 (0.61–1.92)	0.782	13/72	11/159	3.39 (1.35–8.52)	**0.009**
rs7841060	Gender (M)	76/52	28/16	0.93 (0.45–1.92)	0.847	10/118	6/38	0.55 (0.19–1.64)	0.286
Age	Gender (F)	27/30	69/73	0.94 (0.50–1.76)	0.838	2/55	10/132	0.37 (0.07–1.88)	0.230
<60	IGC	43/43	97/89	0.68 (0.36–1.28)	0.232	5/81	16/170	0.43 (0.13–1.37)	0.153
	DGC	47/30	97/89	1.39 (0.77–2.50)	0.272	4/73	16/170	0.48 (0.15–1.57)	0.224
	LNM (−)	71/49	97/89	1.19 (0.68–2.08)	0.535	6/114	16/170	0.34 (0.12–1.03)	0.056
	LNM (+)	32/33	97/89	0.74 (0.39–1.37)	0.333	6/59	16/170	0.86 (0.30–2.47)	0.778
	Tumor stage I + II	82/59	97/89	1.11 (0.66–1.88)	0.695	9/132	16/170	0.49 (0.19–1.29)	0.147
	Tumor stage III	21/23	97/89	0.67 (0.33–1.36)	0.264	3/41	16/170	0.55 (0.14–2.11)	0.384
≥60	Gender (M)	93/83	36/37	1.14 (0.66–1.98)	0.644	20/156	6/67	1.35 (0.51–3.56)	0.544
	Gender (F)	37/39	55/43	0.74 (0.38–1.44)	0.378	6/70	7/91	1.31 (0.39–4.42)	0.668
	IGC	85/73	91/80	1.15 (0.71–1.86)	0.577	15/143	13/158	1.26 (0.53–3.01)	0.602
	DGC	32/31	91/80	0.90 (0.50–1.64)	0.737	7/56	13/158	1.38 (0.50–3.78)	0.536
	LNM (−)	79/67	91/80	1.08 (0.67–1.73)	0.753	14/132	13/158	1.26 (0.54–2.95)	0.598
	LNM (+)	51/55	91/80	0.82 (0.48–1.39)	0.450	12/94	13/158	1.22 (0.48–3.09)	0.676
	Tumor stage I + II	85/81	91/80	0.98 (0.62–1.55)	0.922	16/150	13/158	1.28 (0.56–2.93)	0.554
	Tumor stage III	45/41	91/80	0.95 (0.54–1.67)	0.861	10/76	13/158	1.21 (0.45–3.25)	0.702
**rs16901946**	Gender (M)	66/63	17/27	1.48 (0.72–3.02)	0.282	14/115	3/41	1.64 (0.44–6.05)	0.462
Age	Gender (F)	29/26	68/75	1.35 (0.71–2.57)	0.364	1/54	9/134	0.34 (0.04–2.76)	0.312
<60	IGC	45/40	85/102	1.75 (0.92–3.32)	0.088	10/75	12/175	1.38 (0.46–4.11)	0.562
	DGC	39/38	85/102	1.20 (0.68–2.14)	0.533	2/75	12/175	0.35 (0.07–1.81)	0.210
	LNM (−)	58/62	85/102	1.14 (0.66–1.98)	0.638	10/110	12/175	0.95 (0.33–2.74)	0.920
	LNM (+)	37/27	85/102	1.78 (0.96–3.32)	0.069	5/59	12/175	1.09 (0.33–3.58)	0.883
	Tumor stage I+II	68/73	85/102	1.08 (0.64–1.83)	0.766	12/129	12/175	1.01 (0.37–2.76)	0.989
	**Tumor stage III**	27/16	85/102	2.38 (1.15–4.94)	**0.020**	3/40	12/175	1.03 (0.26–4.08)	0.972
≥60	**Gender (M)**	91/85	40/33	0.91 (0.52–1.58)	0.734	15/161	12/61	0.43 (0.19–0.98)	**0.046**
	Gender (F)	43/34	54/43	1.14 (0.58–2.22)	0.709	8/69	8/89	1.33 (0.41–4.25)	0.635
	**IGC**	82/77	94/76	0.94 (0.58–1.52)	0.797	12/147	20/150	0.38 (0.16–0.89)	**0.026**
	DGC	47/30	94/76	1.43 (0.79–2.56)	0.236	4/73	20/150	0.50 (0.15–1.63)	0.248
	LNM (−)	80/66	94/76	1.05 (0.65–1.69)	0.841	16/130	20/150	0.69 (0.32–1.47)	0.336
	LNM (+)	54/53	94/76	0.85 (0.50–1.44)	0.538	7/100	20/150	0.39 (0.15–1.06)	0.064
	Tumor stage I + II	92/75	94/76	1.04 (0.66–1.65)	0.871	18/149	20/150	0.65 (0.31–1.36)	0.249
	Tumor stage III	42/44	94/76	0.82 (0.47–1.45)	0.495	5/81	20/150	0.37 (0.12–1.12)	0.078

GC, gastric cancer; CON, control; ht, heterozygous; mt, mutant; wt, wild-type; SNP, single nucleotide polymorphism; AOR, adjusted odds ratio; CI, confidence interval; M, male; F, female; IGC, intestinal-type gastric cancer; DGC, diffuse-type gastric cancer; LNM; lymph node metastasis. ^a^ Adjusted for age and gender. The significant results are in bold.

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
