# Peer review of "Correlations between Genetic Polymorphisms in Long Non-Coding RNA PRNCR1 and Gastric Cancer Risk in a Korean Population"

_ijms, 2019, doi:10.3390/ijms20133355_

Round 1
Reviewer 1 Report
In this study, the authors utilized a Taqman genotyping assay to analyze PRNCR1 polymorphisms in gastric cancer patients and controls. Five SNPs of this gene were analyzed and stratification analysis was performed as well. The number of cases and controls is appropriate.
Results: Only one genotype of one SNP was associated with an increased risk for GC. Although significant, the difference between the control and GC group for this SNP (rs13252298) was very small.
11% in GC and 9.4% in control. This should be discussed.
In the stratified analysis, several differences between the control and GC group were noted.
Comments: Tables 2 and 4 contain a great deal of information and the font is small. Is it possible to present the data in several tables? This may take up more space, but the results would be easier to view. In table 3, the genotypes are not presented. They should be added.
Line 55: the authors state “based on previous findings..”, but no references are presented. Why do they think PRNCR1 polymorphisms may be related to GC? Is there a biological basis?
What is known about the gene and the SNPs with respect to how they are related to increased cancer risk? This should be discussed in more detail.
Line 146: What ethical considerations prevented the authors from including H. pylori infection in the analysis?
Line 153: It is correct to state that some polymorphisms are associated with a greater risk. It is not correct to state that “the…polymorphisms increase susceptibility to GC…”
The data presentation is the most important issue.
Author Response
Comments and Suggestions for Authors
In this study, the authors utilized a Taqman genotyping assay to analyze PRNCR1 polymorphisms in gastric cancer patients and controls. Five SNPs of this gene were analyzed and stratification analysis was performed as well. The number of cases and controls is appropriate.
Results: Only one genotype of one SNP was associated with an increased risk for GC. Although significant, the difference between the control and GC group for this SNP (rs13252298) was very small.
11% in GC and 9.4% in control. This should be discussed.
In the stratified analysis, several differences between the control and GC group were noted.
Comments: Tables 2 and 4 contain a great deal of information and the font is small. Is it possible to present the data in several tables? This may take up more space, but the results would be easier to view. In table 3, the genotypes are not presented. They should be added.
Response 1: According to reviewer’s comments, we change the font to a larger one in Table 2 and 3. Each SNP has different genotype, so it is difficult to present all genotypes in one table.
Line 55: the authors state “based on previous findings..”, but no references are presented. Why do they think PRNCR1 polymorphisms may be related to GC? Is there a biological basis?
Response 2: According to reference 25 and 26, they showed a significant association between PRNCR1 polymorphisms (rs13252298, rs16901946, and rs1456315) and GC risk in Chinese population. Base on their studies, we attempted case-control study in Korean population.
What is known about the gene and the SNPs with respect to how they are related to increased cancer risk? This should be discussed in more detail.
Response 3: As you know, there have been a number of reports that a SNP located in gene regulatory or gene cording region affects a gene (including cancer related gene) expression. According to refernce10, they suggest that germline variants can affect lncRNA expression, regulating cancer development, and progression. Therefore, we supposed that a genetic variation in lncRNA (PRNCR1) can affect lncRNA expression, and it may regulate a cancer related gene expression contributing to cancer development.
Line 146: What ethical considerations prevented the authors from including H. pylori infection in the analysis?
Response 4: When we provided sample and their clinical data by the Chungnam National Hospital Biobank, data about H. pylori infection in all of control and several GC was missed. We cannot access data base to get missing data.
Line 153: It is correct to state that some polymorphisms are associated with a greater risk. It is not correct to state that “the…polymorphisms increase susceptibility to GC…”
Response 5: We agree with the reviewer’s comment. We correct that “PRNCR1 rs13252298 and rs16901946 polymorphisms are associated increased GC risk~”
The data presentation is the most important issue.
Thank you for giving us opportunity to make some changes and improve our manuscript.
Reviewer 2 Report
Understanding population-specific genetic underpinnings of complex diseases is a worthy quest which can be used to desig screening tests or enable personalized treatment options. The authors present a case-control study of association of 5 PRNCR1 SNPs with the risk of gastric cancer (GC) in a Korean population. The fundamental logic of the study is sound; however, I have a few methodological concerns with the genetic and statistical aspects of the study.
Major comments:
1. Sex and age: These are considered confounder candidates in most epidemiological studies; but the existence of confounding must be first established. Contrary to somatic mutation or gene expression, it is not obvious that SNP allele frequencies should be associated with age or sex (unless there is differential genotype-dependent mortality). Establishing confounding effects, therefore, requires demonstrating two facts: A. the potential confounder i.e. age or sex, is significantly associated with both main variables of interest i.e. SNP allele frequencies AND GC risk. B. Including the confounder in the analysis changes the effect size (odds ratio, regression coefficient or any other similar metric depending on the analysis method) by 20% or more. If both conditions are met, one has established confounding and can start thinking about the proper way of handling it. It can be done with confounder-adjusted models, stratified analysis, sampling matched case-control groups, etc. If confounding is not established, the analysis should be performed without sample partitioning or model adjustment. This has not been done here. Stratified or adjusted analysis should be done when warranted and not merely as a tool in search of smaller p-values.
2. P-values must be corrected for multiple testing. Although there is no consensus among statisticians on how to correct across multiple models, correction must be done at least within each model.
3. The 5 SNPs tested here are on the same gene. They must be checked for LD before each is tested for association. If they are strongly linked, it will be impossible to confidently ascertain association for any single SNP. PCA or other appropriate methods may be used. If they are not linked in the study sample, each SNP may be analyzed separately (as done here).
4. The rationale in excluding SNPs deviating from HWE in GWAS is that such SNPs usually represent genotyping errors or interference of structural variations in calling SNPs. But, deviation from HWE can be caused by true association with the diseases, too (for example see Turner S, Armstrong LL, Bradford Y, et al. Quality control procedures for genome-wide association studies. Curr Protoc Hum Genet. 2011 or Marees AT, de Kluiver H, Stringer S, et al. A tutorial on conducting genome-wide association studies: Quality control and statistical analysis. Int J Methods Psychiatr Res. 2018). Since genotyping error and structural variation are not likely here, it is not necessary to exclude rs1456315. Suffices to provide the results of HWE test separately in test and control groups for this SNP, and its deviation from HWE mentioned in the tables’ captions.
5. The individuals participating in genetic case-control studies must be unrelated to avoid bias. Relatedness (or lack of it) cannot be reliably ascertained through 5 closely linked loci; so, it must be established during sample collection by asking for pedigree information and ensuring that individuals do not share ancestry more recently than 4 generations. Was any precaution taken to avoid or correct for relatedness?
Minor comments:
1. L 41: Please specify the time period.
2. L 45: Change “are correlated with” >> “are associated with” or “reside on”. “Correlation” describes the relationship of numerical variables.
3. Please provide references for each one of the 5 chosen SNPs.
4. Please explain how the cutoff of 60y was arrived at to dichotomize age.
5. Discussion: please summarize the significant associations found in this study in a table and compare each with corresponding ones in previous reports. Especially, point out whether the direction of effect (GC risk increase or decrease) is similar or opposite.
Author Response
Comments and Suggestions for Authors
Understanding population-specific genetic underpinnings of complex diseases is a worthy quest which can be used to desig screening tests or enable personalized treatment options. The authors present a case-control study of association of 5 PRNCR1 SNPs with the risk of gastric cancer (GC) in a Korean population. The fundamental logic of the study is sound; however, I have a few methodological concerns with the genetic and statistical aspects of the study.
Major comments:
1. Sex and age: These are considered confounder candidates in most epidemiological studies; but the existence of confounding must be first established. Contrary to somatic mutation or gene expression, it is not obvious that SNP allele frequencies should be associated with age or sex (unless there is differential genotype-dependent mortality). Establishing confounding effects, therefore, requires demonstrating two facts: A. the potential confounder i.e. age or sex, is significantly associated with both main variables of interest i.e. SNP allele frequencies AND GC risk. B. Including the confounder in the analysis changes the effect size (odds ratio, regression coefficient or any other similar metric depending on the analysis method) by 20% or more. If both conditions are met, one has established confounding and can start thinking about the proper way of handling it. It can be done with confounder-adjusted models, stratified analysis, sampling matched case-control groups, etc. If confounding is not established, the analysis should be performed without sample partitioning or model adjustment. This has not been done here. Stratified or adjusted analysis should be done when warranted and not merely as a tool in search of smaller p-values.
Response 1: In Table, distribution of age and gender was significant different between case and control group. Therefore, we used binary logistic regression with adjustment for age and gender to reduce the effect of covariate in this study as other studies.
- Niu F, Wang T, Li J, Yan M, Li D, Li B, Jin T. The impact of genetic variants in IL1R2 on cervical cancer risk among Uygur females from China: A case-control study. Mol Genet Genomic Med. 2019 Jan;7(1):e00516.
- Wang C, Zhang C, Xu J, Li Y, Wang J, Liu H, Liu Y, Chen Z, Lin H. Association between IL-1R2 polymorphisms and lung cancer risk in the Chinese Han population: A case-control study. Mol Genet Genomic Med. 2019 May;7(5):e644.
- Wei L, Niu F, Wu J, Chen F, Yang H, Li J, Jin T, Wu Y. Association study between genetic polymorphisms in folate metabolism and gastric cancersusceptibility in Chinese Han population: A case-control study. Mol Genet Genomic Med. 2019 May;7(5):e633.
- He BS, Sun HL, Xu T, Pan YQ, Lin K, Gao TY, Zhang ZY, Wang SK. Association of Genetic Polymorphisms in the LncRNAs with Gastric Cancer Risk in a Chinese Population. J Cancer. 2017 Feb 11;8(4):531-536.
2. P-values must be corrected for multiple testing. Although there is no consensus among statisticians on how to correct across multiple models, correction must be done at least within each model.
Response 2: Our study is a kind of exploratory study and we didn’t correct for multiple testing as other case-control studies.
3. The 5 SNPs tested here are on the same gene. They must be checked for LD before each is tested for association. If they are strongly linked, it will be impossible to confidently ascertain association for any single SNP. PCA or other appropriate methods may be used. If they are not linked in the study sample, each SNP may be analyzed separately (as done here).
Response 3: We estimated LD coefficients (|D’|) among the four SNPs, and an absolute LD (|D’|=1 and r2) was not found for any pair-wise combination among four SNPs using Haploview 4.0 software. We added this results in manuscript (L80-82 and L182-184).
4. The rationale in excluding SNPs deviating from HWE in GWAS is that such SNPs usually represent genotyping errors or interference of structural variations in calling SNPs. But, deviation from HWE can be caused by true association with the diseases, too (for example see Turner S, Armstrong LL, Bradford Y, et al. Quality control procedures for genome-wide association studies. Curr Protoc Hum Genet. 2011 or Marees AT, de Kluiver H, Stringer S, et al. A tutorial on conducting genome-wide association studies: Quality control and statistical analysis. Int J Methods Psychiatr Res. 2018). Since genotyping error and structural variation are not likely here, it is not necessary to exclude rs1456315. Suffices to provide the results of HWE test separately in test and control groups for this SNP, and its deviation from HWE mentioned in the tables’ captions.
Response 4: Deviation from HEW derived from genotyping error or evolutional forces (ex) inbreeding, migration, genetic draft etc.) In case of evolutional forces, it cannot be cause of disease. Therefore, we excluded rs1456315 in data analysis.
5. The individuals participating in genetic case-control studies must be unrelated to avoid bias. Relatedness (or lack of it) cannot be reliably ascertained through 5 closely linked loci; so, it must be established during sample collection by asking for pedigree information and ensuring that individuals do not share ancestry more recently than 4 generations. Was any precaution taken to avoid or correct for relatedness?
Response 5: Thank you for your suggestion. Unfortunately, we could not correct for relatedness because it is impossible to collect pedigree information when we provided sample from a Biobank.
Minor comments:
1. L 41: Please specify the time period.
Response 6: According to reviewer’s comment, we represent a year in L41.
2. L 45: Change “are correlated with” >> “are associated with” or “reside on”. “Correlation” describes the relationship of numerical variables.
Response 7: We have corrected the manuscript as suggested.
3. Please provide references for each one of the 5 chosen SNPs.
Response 8: According to reviewer’s comment, we added references in L177 and 178.
4. Please explain how the cutoff of 60y was arrived at to dichotomize age.
Response 9: We cutoff age with references to previous reports.
- Gu D, Wang M, Wang S, Zhang Z, Chen J. The DNA repair gene APE1 T1349G polymorphism and risk of gastric cancer in a Chinesepopulation. PLoS One. 2011;6(12):e28971.
- Zhang H, Jin G, Li H, Ren C, Ding Y, Zhang Q, Deng B, Wang J, Hu Z, Xu Y, Shen H. Genetic variants at 1q22 and 10q23 reproducibly associated with gastric cancer susceptibility in a Chinese population. Carcinogenesis. 2011 Jun;32(6):848-52.
- Zhao Q, Wang W, Zhang Z, Wang S, Wang M, Zhou J, Gong W, Tan Y, Wang B, Chen G. A genetic variation in APE1 is associated with gastric cancer survival in a Chinese population. Cancer Sci. 2011 Jul;102(7):1293-7.
- Cao X, Zhuang S, Hu Y, Xi L, Deng L, Sheng H, Shen W. Associations between polymorphisms of long non-coding RNA MEG3 and risk of colorectal cancer in Chinese. Oncotarget. 2016 Apr 5;7(14):19054-9.
- Ma X, Huang C, Luo D, Wang Y, Tang R, Huan X, Zhu Y, Xu Z, Liu P, Yang L. Tag SNPs of long non-coding RNA TINCR affect the genetic susceptibility to gastric cancer in a Chinese population. Oncotarget. 2016 Dec 27;7(52):87114-87123.
- Zou JH, Li CY, Bao J, Zheng GQ. High expression of long noncoding RNA Sox2ot is associated with the aggressive progressionand poor outcome of gastric cancer. Eur Rev Med Pharmacol Sci. 2016 Nov;20(21):4482-4486.
- Ge Y, He Y, Jiang M, Luo D, Huan X, Wang W, Zhang D, Yang L, Zhou J. Polymorphisms in lncRNA PTENP1 and the Risk of Gastric Cancer in a Chinese Population. Dis Markers. 2017;2017:6807452.
- He BS, Sun HL, Xu T, Pan YQ, Lin K, Gao TY, Zhang ZY, Wang SK. Association of Genetic Polymorphisms in the LncRNAs with Gastric Cancer Risk in a ChinesePopulation. J Cancer. 2017 Feb 11;8(4):531-536.
- Lv Z, Sun L, Xu Q, Gong Y, Jing J, Xing C, Yuan Y. Long non-coding RNA polymorphisms in 6p21.1 are associated with atrophic gastritis risk and gastric cancer prognosis. Oncotarget. 2017 Aug 10;8(56):95303-95315.
5. Discussion: please summarize the significant associations found in this study in a table and compare each with corresponding ones in previous reports. Especially, point out whether the direction of effect (GC risk increase or decrease) is similar or opposite.
Response 10: We represented in L131-140.
Thank you for giving us opportunity to make some changes and improve our manuscript.
Reviewer 3 Report
In this manuscript, the authors assess the association between a long non-coding RNA, PRNCR1, and gastric cancer in the Korean population. The authors present evidence that certain genetic variations in PRNCR1 may slightly enhance susceptibility to gastric cancer. The manuscript is well written and is acceptable for publication provided the following concerns are addressed.
Major Comment
The sample size is very small for both the GC and Control groups and is limited to a small ethnic group in Korea. Can the authors at least increase the sample size to increase the statistical power particularly for the stratified analysis?
Minor Comments
The authors should include “in a Korean population” in the title as the study was limited to a small GC subgroup in Korea.
In the Introduction Line 45, it is mentioned that LncRNAs are non-transcribed. Did the authors mean to say “non-translated”? This must be corrected.
More detail on the TaqMan SNP Genotyping assay should be provided. Also, the authors must include the list of primers used for this assay for all genotypes tested in a supplement.
The Tables 2 and 3 are quite complicated in their presentation and could be simplified either by including only the strongest of SNPs and/or by highlighting the SNPs and entries that are suggestive of a strong phenotype.
In the discussion section, the authors should include a paragraph on the putative mechanism of PRNCR1 role in GC based on previously published literature and conjecture on how certain polymorphisms may contribute to a decreased risk of GC in<60 age group.
The formatting of References is not appropriate and must be fixed.
Author Response
Comments and Suggestions for Authors
In this manuscript, the authors assess the association between a long non-coding RNA, PRNCR1, and gastric cancer in the Korean population. The authors present evidence that certain genetic variations in PRNCR1 may slightly enhance susceptibility to gastric cancer. The manuscript is well written and is acceptable for publication provided the following concerns are addressed.
Major Comment
The sample size is very small for both the GC and Control groups and is limited to a small ethnic group in Korea. Can the authors at least increase the sample size to increase the statistical power particularly for the stratified analysis?
Response 1: Thank you for your suggestion. Unfortunately, we could not perform further investigation because of IRB approval and research fund.
Minor Comments
The authors should include “in a Korean population” in the title as the study was limited to a small GC subgroup in Korea.
Response 2: We have corrected the title as suggested.
In the Introduction Line 45, it is mentioned that LncRNAs are non-transcribed. Did the authors mean to say “non-translated”? This must be corrected.
Response 3: We agree with the reviewer about that. We corrected.
More detail on the TaqMan SNP Genotyping assay should be provided. Also, the authors must include the list of primers used for this assay for all genotypes tested in a supplement.
Response 4: Thank you for your suggestion. Unfortunately, we genotyped using the Applied Biosystems TaqMan SNP Genotyping Assay and they do not supply a sequence.
The Tables 2 and 3 are quite complicated in their presentation and could be simplified either by including only the strongest of SNPs and/or by highlighting the SNPs and entries that are suggestive of a strong phenotype.
Response 5: According to reviewer’s comments, we change the font to a larger one and highlight the SNPs showing association in bold.
In the discussion section, the authors should include a paragraph on the putative mechanism of PRNCR1 role in GC based on previously published literature and conjecture on how certain polymorphisms may contribute to a decreased risk of GC in<60 age group.
Response 6: Thank you for your suggestion. There have been a number of reports that a SNP located in gene regulatory or gene cording region affects a gene (including cancer related gene) expression. According to reference10, they suggest that germline variants can affect lncRNA expression, regulating cancer development, and progression. Unfortunately, we just suggest that rs1016343 is associated with protective risk of GC of <60 age group without lymph node metastasis in recessive model. However, we couldn’t demonstrate how the rs1016343 decrease the risk of GC of <60 aged group without lymph node metastasis without any biological evedence. We can hypothesis cautiously that rs1016343 variation may affect PRNCR1 expression and lead to protection for cancer development by up- regulating a tumor suppressor gene or down-regulating a cancerous gene.
The formatting of References is not appropriate and must be fixed.
Response 7: According to reviewer’s comments, we formatted references.
Thank you for giving us opportunity to make some changes and improve our manuscript.
Round 2
Reviewer 2 Report
Please see attached file.
Reviewer 3 Report
The authors have addressed all the concerns appropriately and the manuscript is now acceptable for publication.